# The Bovine Seminal Plasma Protein PDC-109 Possesses Pan-Antiviral Activity

**DOI:** 10.3390/v14092031

**Published:** 2022-09-13

**Authors:** Hannah Sabeth Sperber, Kathrin Sutter, Karin Müller, Peter Müller, Roland Schwarzer

**Affiliations:** 1Institute for Translational HIV Research, University Hospital Essen, Hufelandstraße 55, 45147 Essen, Germany; 2Institute for Virology, University Hospital Essen, Virchowstr. 171, 45147 Essen, Germany; 3Leibniz Institute for Zoo and Wildlife Research, Alfred-Kowalke-Straße 17, 10315 Berlin, Germany; 4Department of Biology, Humboldt-University Berlin, Invalidenstr. 42, 10115 Berlin, Germany

**Keywords:** SARS-CoV-2, VSV replicon, PDC-109, Bovine seminal plasma, Fn-type 2 proteins

## Abstract

Mammalian seminal plasma contains a multitude of bioactive components, including lipids, glucose, mineral elements, metabolites, proteins, cytokines, and growth factors, with various functions during insemination and fertilization. The seminal plasma protein PDC-109 is one of the major soluble components of the bovine ejaculate and is crucially important for sperm motility, capacitation, and acrosome reaction. A hitherto underappreciated function of seminal plasma is its anti-microbial and antiviral activity, which may limit the sexual transmission of infectious diseases during intercourse. We have recently discovered that PDC-109 inhibits the membrane fusion activity of influenza virus particles and significantly impairs viral infections at micromolar concentrations. Here we investigated whether the antiviral activity of PDC-109 is restricted to Influenza or if other mammalian viruses are similarly affected. We focused on Severe Acute Respiratory Syndrome Coronavirus-2 (SARS-CoV-2), the etiological agent of the Coronavirus Disease 19 (COVID-19), thoroughly assessing PDC-109 inhibition with SARS-CoV-2 Spike (S)-pseudotyped reporter virus particles, but also live-virus infections. Consistent with our previous publications, we found significant virus inhibition, albeit accompanied by substantial cytotoxicity. However, using time-of-addition experiments we discovered a treatment regimen that enables virus suppression without affecting cell viability. We furthermore demonstrated that PDC-109 is also able to impair infections mediated by the VSV glycoprotein (VSVg), thus indicating a broad pan-antiviral activity against multiple virus species and families.

## 1. Introduction

One major route of infection for both viral and bacterial pathogens is sexual contact. Infections can induce an insurmountable number of sexually transmitted diseases (STDs), causing severe, often life-threatening pathologies and large-scale epidemics within human populations. The abundance of viruses in male ejaculates and seminal plasma has severe consequences for the male and, after sexual transmission, the recipient sexual partner [1], and it is reasonable to assume that hosts have evolved ample mechanisms to cope with this threat. In search of novel antiviral substances, we have focused our present study on mammalian seminal plasma (SP), which has been previously shown to possess antiviral activities [2,3]. Interestingly, SP contains both pro- and antiviral agents, amongst them differently acting hormones, cytokines, and growth factors. Virus infection-promoting factors include amyloid fibrils [4] (which are aggregated fragments of the prostatic acid phosphatase), substances neutralizing the vaginal acidic pH [5], and complement fragments enhancing virus–target cell attachment [6]. In contrast, proteins such as defensins, lactoferrin, and clusterin exert antiviral activity, which is likely to protect spermatozoa against viral contaminations within the male and female genital tract [7,8]. Moreover, human seminal exosomes have been shown to possess anti-HIV-1 activity [9,10,11,12,13].

Recently, we have found that the SP protein PDC-109 (also known as BSP-1/2) is able to inhibit the infectivity of the influenza virus [14]. PDC-109 is the major component of bovine SP and belongs to a protein family containing the fibronectin type II domain (Fn II). Fn II proteins were found in SP of many other mammalian species, including humans [15]. A couple of previous studies have characterized PDC-109, its molecular interactions, and its physiological activities/role in detail [15,16,17,18,19,20]. Briefly, PDC-109 specifically interacts with choline-containing phospholipids of (sperm) membranes and, thereby, modulates membrane and cell properties [21,22,23,24]. These alterations are part of a cascade of processes that prime the male gametes for fertilization, i.e., stabilizing sperm cells during their transit through the female genital tract, facilitating the formation of the sperm reservoir by mediating sperm-binding to the oviductal epithelium, and subsequent destabilization of the sperm membrane enabling the acrosomal exocytosis prior to fertilization [15,17,20]. Additionally, a chaperone-like activity of PDC-109 has been reported [25]. Recently, we discovered an influence of PDC-109 on influenza virus activity, which is caused by the binding of PDC-109 to both viral and target membranes [14]. Similar to its interaction with sperm cells, binding probably occurs to choline-containing lipids, thereby preventing virus–cell fusion. Based on this mode of action, we hypothesize that PDC-109 might exert broader antiviral effects.

To test this hypothesis, we have assessed the inhibitory activity of PDC-109 on reporter viruses, pseudotyped with either the spike protein of the severe acute respiratory syndrome coronavirus type 2 (SARS-CoV-2) or the Vesicular stomatitis virus glycoprotein (VSVg). By using two non-related virus species, we sought to explore whether the antiviral activity is pathogen-specific or pan-viral. We observed significant, dose-dependent pan-antiviral effects of PDC-109 with only minor cytotoxicity upon the transient administration of PDC-109. Interestingly, extended exposure of mammalian cells to PDC-109 leads to a marked reduction in cell viability through an unknown mechanism.

## 2. Materials and Methods

**Mammalian Cell culture.** The cell lines BHK-G43 (a kind gift from Gert Zimmer, Institute of Virology and Immunology, Mittelhäusern, Switzerland), Vero E6 (CRL-1586; American Type Culture Collection, Manassas, VA, USA), Calu6 (HTB-56; American Type Culture Collection, Manassas, VA, USA), and HEK293T (CRL-3216; American Type Culture Collection, Manassas, VA, USA) were maintained in Dulbecco’s modified Eagle medium (DMEM) containing 10% heat-inactivated fetal bovine serum (FBS), 2 mM L-glutamine, 100 U/mL penicillin, and 100 μg/mL streptomycin (all from PAA Laboratories GmbH, Pasching, Austria) under standard cell culture conditions.

**Chemicals.** If not otherwise stated, all of the chemicals were purchased from Sigma-Aldrich (Taufkirchen, Germany). The phosphate-buffered saline (PBS) contained 150 mM NaCl and 5.8 mM sodium phosphate (pH 7.4). The HEPES-buffered salt solution (HBS) contained 150 mM NaCl and 10 mM HEPES (pH 7.4).

**Seminal plasma.** The semen was collected from bulls (*Bos taurus taurus*) routinely used for the production of doses for artificial insemination (AI) in a breeding center located in Germany. The semen production protocols were applied according to the general guidelines for semen processing used in AI centers. The semen was collected with an artificial vagina and transported to the laboratory facility within 5 min. Only aliquots of ejaculates with sperm motility above 80% were used. The sperm cells were immediately pelleted by centrifugation (1200× *g*, 8 min, RT), and the supernatant was centrifuged again in 2 mL aliquots to remove the residual sperm cells (12,000× *g*, 2 min, RT). SP was aliquoted and frozen (−20 °C) until further processing.

**Purification of PDC-109.** PDC-109 was purified from the delipidated SP (dSP) samples as described before [26]. The delipidation of SP was performed according to [27]. For that, the SP samples were centrifuged (10 min, 10,000× *g*). One volume (vol) of the supernatant was mixed with 9 vol ice-cold EtOH and stirred for 90 min at 4 °C, followed by centrifugation of the solution (10 min, 10,000× *g*). The resultant pellet was washed three times with ice-cold EtOH (centrifugation 10 min, 10,000× *g*) and subsequently resolved in 1 vol 50 mM NH_4_HCO_3_ and finally lyophilized. The resultant powder was solved in a minimal volume of HBS followed by the addition of 2 vol isopropyl ether/n-butanol (60:40) and incubation for 30 min on a shaker. The suspension was centrifuged (2 min, 1000× *g*), and the organic phase containing lipids was removed by aspiration. The residual organic solvent above the water phase was eliminated by passing a gentle stream of nitrogen over the aqueous layer. The final aqueous solution containing delipidated proteins was lyophilized.

For the purification of PDC-109, the freeze-dried sample was solubilized in TBS buffer (50 mM Tris, 1 M NaCl, 5 mM EDTA, pH 6.4) and centrifuged (5 min, 500× *g*). The supernatant was given on a DEAE A25 column (Merck KGaA, Darmstadt, Germany) linked to a Bio-Rad Econo system (Bio-Rad, Feldkirchen, Germany) and flushed with TBS until the protein absorption at 280 nm declined to baseline. Subsequently, PDC-109 was eluted with TBS, additionally containing 100 mM choline chlorid. The fractions containing PDC-109 were combined and extensively dialyzed against 50 mM NH_4_HCO_3_, pH 8.0 using dialysis tubes MEMBRA-CEL^®^, MW 7000 (Serva, Heidelberg, Germany), followed by lyophilization. For the experiments, PDC-109 was diluted in HBS verifying protein concentration by measuring its absorption at 280 nm (using A_280_ of 1 mg/mL = 2.5) [26].

**Establishment of HEK293T and Calu6 cells stably expressing ACE2 and TMPRSS2.** Human ACE2 was amplified from pCG1-hACE2 (a kind gift from Graham Simmons) and cloned into pLKO5d.SFFV.dCas9-KRAB.P2A.BSD (Addgene, Cat.#90332, a gift from Dirk Heckl). Human TMPRSS2 was amplified from pQCXIBL-hTMPRSS2 (a kind gift from Graham Simmons) and cloned into pDUAL CLDN (GFP) (Addgene, Cat.#86981, a gift from Joe Grove) together with a Puromycin resistance gene, replacing Claudin and GFP, respectively. Clonings were confirmed by Sanger sequencing. Lentiviral particles for lentiviral transfer of hACE2 and hTMPRSS2 were produced in HEK293T cells using transfection with polyethylenimine (PEI). Briefly, the cells were triple transfected with the lentiviral ACE2 or TMPRSS2 constructs, psPAX2 (Addgene, Cat.#12260, a gift from Didier Trono), and pVSV-G (Addgene, Cat.# #138479, a gift from Akitsu Hotta). Then, 48 h post-transfection, the cell supernatants containing the newly produced viral particles were centrifuged, filtered using 0.22 µm vacuum filter units (Merck Millipore, Darmstadt, Germany) and stored at −80 °C. To establish stable cell lines, the HEK293T or Calu6 cells were first transduced with lentiviral particles containing the ACE2 vector. Then, 48 h post-transduction, the medium was replaced with blastidicin (BSD; InvivoGen, San Diego, CA, USA, 1 μg/mL) selection medium. For the double-transduced cells, after the sufficient selection and expansion of transgenic cells, another round of transduction with TMPRSS2 lentiviral particles followed. The double-transduced cells were selected in an antibiotic selection medium containing BSD and Puromycin (1 μg/mL) 48 h post-transduction. The expression of ACE2 and TMPRSS2 was confirmed by Western Blot.

**VSV*∆G-fLuc pseudotyping and transduction experiments.** The preparation of VSV pseudotyping has been described previously [28]. Briefly, the Hek293T cells were transfected using PEI with a SARS-CoV-2-Spike-encoding plasmid (pCG1-SARS-CoV-2-Spike, kindly provided by Dr. Graham Simmons, Vitalant Research Institut, described in [29], NCBI Reference Sequence: YP_009724390.1) for 24 h and subsequently infected with a single-round VSV reporter replicon (VSV*∆G-fLuc, kindly provided by Gert Zimmer, Institute of Virology and Immunology, Mittelhäusern, Switzerland), in which the VSVg open reading frame was replaced with an enhanced green fluorescent protein (eGFP) and a firefly luciferase.

The cells were incubated for 4 h with the input virus at 37 °C, 5% CO_2_, then washed with PBS before medium, supplemented with anti-VSV-G antibody from Vi-10 hybridoma cells [30] was added in order to neutralize seed virus residues. The next day, the supernatants containing the pseudotyped virus particles were harvested, cleared from cellular debris by centrifugation, filtered using a Syringe Filter (0.22 µm, Sartorius, Göttingen, Germany), and stored at −80 °C.

VSV*∆G-fLuc VSVg seed virus propagation was conducted in BHK-G43 cells as described previously [31]. The cells were pre-plated, and VSVg expression was induced with 1 nM Mifepristone (Calbiochem, Darmstadt, Germany) for 6 h at 37 °C, 5% CO_2_, followed by infections with VSV*∆G-fLuc VSVg (MOI~1) for 16–24 h at 37 °C, 5% CO_2_. Ultimately, supernatants were harvested, cleared from cellular debris by centrifugation, filtered using a Syringe Filter (0.22 µm, Sartorius, Göttingen, Germany), and stored at −80 °C.

For transduction experiments, 25,000–50,000 target cells were plated in 96-well flat-bottom dishes and inoculated the next day with the pseudotyped VSV*∆G-fLuc at an MOI of around 0.1. If not otherwise stated, the cells were infected by spinoculation for 30 min, at 1200× *g* and 37 °C, followed by another 90 min of incubation at 37 °C, 5% CO_2_. Subsequently, the virus-containing supernatant was removed, and the cells were further incubated for 24 h at 37 °C, 5% CO_2_. The infection levels were assessed using flow cytometry and LSRII (Becton Dickinson, San Diego, CA, USA), equipped with a high-throughput sampling unit (HTS), gating on single, live GFP positive cells. If not otherwise stated, PDC-109 and SP were administered 15–30 min prior to and during infections (until supernatant removal after 2 h). Cell viability was assessed by forward and side scatter (FSC and SSC) gating based on gate calibration using Zombie fixable viability dyes (BioLegend, San Diego, CA, USA).

**SARS-CoV-2 propagation, titration and TCID50 experiments.** The SARS-CoV-2 isolate (Essen isolate) used in this study was obtained from patient material and propagated in VeroE6 cells, as previously described [32]. Briefly, 2 × 10^6^ VeroE6 cells were seeded in a T75 flask and maintained in Dulbecco’s modified Eagle’s medium (DMEM) supplemented with 10% fetal bovine serum (FBS), L-glutamine, penicillin, and streptomycin for 24 h at 37 °C, 5% CO_2_. Then, the cells were infected with the isolated virus and cultured for another 72 h. Finally, the supernatant was harvested, cell debris was removed by centrifugation, and the supernatant aliquots were stored at −80 °C. Viral titers were determined by endpoint dilution assay in order to calculate the 50% tissue culture infective dose (TCID50). SARS-CoV-2 inhibition was tested by TCID50 assays using suited inoculums of virus and titrations of antiviral compounds. The cells were pre-treated for 15–30 min with antiviral reagents and subsequently infected in their presence for 2 h at 37 °C, 5% CO_2_. Then, the virus-containing supernatants were removed, and the cells were incubated for another 72 h. Finally, crystal-violet staining was performed in order to identify inhibition of viral cytopathic effects. 8–10 wells were typically assessed per drug concentration, and the fraction of wells without viral plaques in otherwise confluent cell monolayers was calculated as a measure of SARS-CoV-2 infection.

**SARS-CoV-2 In-Cell ELISA.** Virus quantification by In-Cell ELISA was performed as recently described [33]. Briefly, 5 × 10^4^ cells/well (flat-bottom 96-well plate) were plated one-day pre-infection. Then, the cells were infected with SARS-CoV-2 for 16–24 h and fixed with 4% (*w*/*v*) paraformaldehyde/PBS. Then, 1% (*v*/*v*) Triton-X-100/PBS was used for permeabilization, followed by blocking with 3% (*v*/*v*) FCS/PBS. Subsequently, the primary antibody (anti-N mAb1 ABIN6952435, Antibodies Online, Aachen, Germany) was added and incubated for 2 h at room temperature or overnight at 4 °C. Peroxidase-labelled secondary antibody (Cat.#115-035-003, Jackson Immuno Research, Cambridge, UK) was added for another 1–2 h prior to washing steps with 0.05% (*v*/*v*) Tween-20/PBS. Finally, Tetramethylbenzidin (TMB) substrate was added, and the enzymatic reaction was stopped with 0.5 M HCl. The absorbance of the dye was measured at 450 nm using a Spark 10M multimode microplate reader (Tecan, Maennedorf, Switzerland).

**Statistics.** If not stated otherwise, the bars show arithmetric mean ± SEM. Statistical significance was assessed using Prism (GraphPad Software Inc., San Diego, CA, USA), applying parametric one-way analysis of variance (ANOVA) tests and displayed as follows: **** *p* < 0.0001; *** *p* < 0.001; ** *p* = 0.001–0.01; * *p* = 0.01–0.05. The data were tested for normality by Shapiro–Wilk test using a significance level of 0.05.

## 3. Results

We have recently demonstrated that PDC-109 possesses antiviral properties, effectively blocking influenza virus fusion with plasma membranes of susceptible red blood cells [14]. Here, we sought to test whether PDC-109 is also capable of inhibiting the activity of viruses other than influenza.

**PDC-109 effectively blocks infections with SARS-CoV-2 pseudotypes.** Initially, we tested if PDC-109 is able to inhibit infections with SARS-CoV-2 Spike-pseudotyped VSV* (VSV*SARS CoV-2). As a cellular target, we utilized Calu6 ACE2 cells, a transgenic cell line that was transduced with human angiotensin-converting enzyme 2 (ACE2, Appendix A). Calu6 ACE2 cells stably express ACE2, the canonical receptor of SARS-CoV-2 (Appendix A), rendering this lung epithelial cell line permissive for SARS-CoV-2 pseudotypes. The parental cell line, on the other hand, is refractive to SARS-CoV-2 infections, which enables effective testing of SARS-CoV-2 pseudotypes for ACE2 specificity (Appendix A). We first incubated the Calu6 ACE2 cells with increasing concentrations of PDC-109, followed by infection with VSV*SARS CoV-2 for 24 h (Figure 1A). Strong, dose-dependent inhibition of infection was found at higher micromolar concentrations (25–100 μM). Notably, we also observed the significant cytotoxicity of PDC-109 at concentrations of >25 μM (Figure 1A). However, by decreasing the PDC-109 of virus infection, with negligible cytotoxic effects (Figure 1B).

**PDC-109 interferes with SARS-CoV-2 mediated entry and reporter virus replication.** The results described above indicate that PDC-109 has substantial inhibitory effects on virus infections mediated by SARS-CoV-2 spike proteins. Next, we investigated whether PDC-109 also impairs other steps of viral replication cycles. For that, a series of time-of-addition experiments were performed with varying PDC-109 administration intervals (Figure 2A). The cells were exposed to PDC-109 either (i) for the entire 24 h virus replication period, (ii) for the first 2 h of virus entry only, or (iii) 2 h after initiating virus infection. Interestingly, PDC-109 reduced infection levels in all experimental setups (Figure 2B), including post-entry administration (2–24 h). This indicates multifactorial effects of PDC-109 on both SARS-CoV-2-mediated virus entry and VSV* dominated replication and GFP expression. Again, we observed significant toxicity throughout all PDC-109 treated samples (albeit of negligible extent for 2 h treatments), which may confound the potential, antiviral effects at longer administration periods.

**Cells are not protected from SARS-CoV-2 Spike-mediated entry after PDC-109 pre-treatment**. PDC-109 has been shown to bind to membranes modulating the structure and dynamics of membrane lipids [21,22,23,24]. Thus, we speculated whether the impact of PDC-109 on virus infectivity is mediated by modifications of cellular membranes. In that case, PDC-109 may induce irreversible changes that could render cellular membranes non-permissive to viral infections even after its removal from cell culture media. To test this hypothesis, we utilized two transgenic sub-cell lines with different permissivities for SARS-CoV-2 mediated entry. Both 293T ACE2 and 293T ACE2 TMPRSS2 were derived from 293T cells that have been transduced with ACE2 and TMPRSS2, respectively. TMPRSS2 is an entry co-factor for CoV-2 infections [29] and strongly facilitates spike-mediated membrane fusion (Figure 3A,B, see “no PDC” samples). These two cell lines were utilized in order to investigate whether only a pre-treatment of cells with PDC-109 affects moderately (293T ACE2) and highly permissive (293T ACE2 TMPRSS2) cells differently. Noteworthily, neither cell line was protected from VSV*SARS CoV-2 infection upon PDC-109 priming (Figure 3A), whereas infections in the presence of PDC-109 were again heavily decreased (Figure 3B). This finding indicates that PDC-109 effects on virus infectivity are transient and rapidly wane after its removal from treated cells. We have also tested if a virus pre-incubation with PDC-109 further promotes its antiviral effects as compared to a pre-treatment of the infected cells. Again, antiviral activity was observed at concentrations > 20 μM, but no significant difference was found in either pre-treatment regimen (Appendix A).

**PDC-109 reduces SARS-CoV-2 infection in cell culture**. Finally, we assessed whether PDC-109 affects live SARS-CoV-2 infections in cell culture. For that, TCID50 assays were employed, and the antiviral effects were determined at 72 h of infection in the presence of increasing concentrations of PDC-109. In agreement with our pseudovirus experiments, the treated cells showed protection from CoV-2 infections up to levels of around 50% at concentrations above 20 μM (Figure 4A). We note that in this experimental setup, PDC-109 was only present during the first two hours of infection, whereas spreading infections of the wild-type virus are permitted for several days. By that, secondary rounds of infections can be presumed not to be affected by the initial PDC-109 treatment. This latter limitation strongly decreases the maximal antiviral effect to be expected, and we hypothesized that the observed decrease in infection levels is, in fact, indicative of a more pronounced inhibition of viral entry in the presence of PDC-109. To test this hypothesis, we employed an alternative assay, an in-cell ELISA, which can detect viral proteins in infected cells already 24 h post-infection. Our data show the almost complete inhibition of viral infection at the highest PDC-109 concentration, which supports the notion that PDC-109 possesses strong antiviral effects against SARS-CoV-2 (Figure 5B). Importantly, we did not find noticeable toxicity effects of a 2 h PDC-109 treatment in the absence of viral infection (Appendix A).

**PDC-109 inhibits entry of VSV into VeroE6 cells.** Our previous experiments have unequivocally shown that PDC-109 has the potential to inhibit SARS-CoV-2 Spike-mediated entry in different permissive cell lines and can even alleviate viral cytopathic effects in authentic live-virus infection assays. We now sought to investigate whether its antiviral activity is limited to Influenza [14] and SARS-CoV-2 (Figure 1, Figure 2, Figure 3 and Figure 4) or whether other virus pseudotypes would be equally blocked. Therefore, we tested VSV* seed particles, which are essentially VSV*ΔG(Luc) replicons, pseudotyped with the amphotropic VSV glycoprotein (VSVg). We used these particles, from here on forward, called VSV*VSVg, to infect VeroE6 cells in the presence of increasing concentrations of PDC-109 for two hours, followed by the removal of the virus-containing supernatent and PDC-109, respectively. A marked and dose-dependent reduction of the GFP+ cells was found upon PDC-109 treatment, with overall maximum inhibition levels of up to 75% (Figure 5A,B). Notably, the delipidated bull SP (dSP) had a similar impact (Figure 5A,B). PDC-109 is derived from dSP, which was included to assess whether antiviral effects are also exerted in the context of complete soluble fractions of ejaculate preparations. Again, the cell viabilities were barely affected by either the PDC-109 or dSP treatment, which even seemed to slightly increase cell survival at high concentrations (Figure 5C). This phenomenon is likely a result of the protection from VSV*VSVg infections since this amphotropic pseudotype leads to very strong infections with severe cytopathic effects. Notably, the post-entry effects of PDC-109 we observed against VSV*SARS CoV-2 (Figure 2) were also apparent in time-of-addition experiments with VSV*SARS VSVg (Appendix A), demonstrating that PDC-109 effectively blocks virus replication through and yet unknown mechanism.

## 4. Discussion

The bioactive components of mammalian SP exhibit a multitude of functions, such as the maintenance of sperm homeostasis or the control of fertilization processes [18,34,35]. In addition, SP has important antimicrobial and antiviral properties, which are so far insufficiently understood. We have recently discovered that the bull SP protein PDC-109 effectively inhibits membrane fusion between influenza virus and red blood cells [14]. We now extend our study to other viruses to better understand this putative physiological function of PDC-109, which could inform the development of novel bio-inspired and biomimetic drugs.

Many viruses require strict biosafety measures and are often limited to BSL3 facilities, with species-specific approval of experimental procedures by governmental authorities. An attractive alternative is pseudotyped, recombinant reporter viruses, such as the VSV*ΔG(Luc) replicon system, developed by Zimmer and colleagues [28]. Multiple studies have shown that pseudotyped VSV*ΔG(Luc) faithfully replicate the entry requirements and other pathogen-specific cellular processes of the wild-type viruses the pseudotyping protein was derived from [36,37,38,39,40,41]. In light of the ongoing SARS-CoV-2 pandemic, we first tested whether PDC-109 could have protective effects against VSV* entry, mediated by the SARS-CoV-2 spike protein. In recent months, the VSV*ΔG(Luc) replicon and related systems have been extensively used for SARS-CoV-2 virus entry research, as well as numerous broad-scale neutralization and seroprevalence studies [29,42,43,44,45,46]. Here, we show that PDC-109 completely abolishes VSV*SARS CoV-2 infections at high micromolar concentrations; however, it is accompanied by significant cytotoxicity when applied for extended periods of time (24 h, Figure 1). However, if the PDC-109 treatment is transient (limited to 2 h), it permits the effective suppression of viral entry with almost no detectable impairment of cellular integrity (Figure 1B and Figure 2). Curiously though, PDC-109 also impacted VSV*SARS CoV-2 infections when it was added to the cells after virus entry (Figure 2 and Appendix A), suggesting that VSV* replication too is subject to PDC-109 antiviral activity. Interestingly, pre-treatment and priming with PDC-109 did not protect the cells from VSV*SARS CoV-2 infections (Figure 3), indicating that PDC-109 exerts antiviral effects directly and only when present during viral infections. Our experiments also revealed a certain degree of cell-line dependent toxicity of PCD-109, with VeroE6 cells not being negatively affected even by 22 h of exposure (Appendix A), whereas the 293T ACE2 cells exhibited minor but significant viability changes already upon 15 min treatment only. This finding could reflect differences in the overall stress resistance and robustness of the respective cell lines. However, it may also indicate that cellular factors being differentially expressed across these cell lines are mediating or alleviating the toxic effects. In this context, the observed differences between 293T ACE2 TMPRSS2 and 293T ACE2 are of particular interest (Figure 3) because they could suggest that the serine protease TMPRSS2 reduces the toxic effects of PDC-109 without affecting its antiviral activity.

We also tested if PDC-109 could alleviate the infection burden in live infections with fully infectious SARS-CoV-2 particles (Figure 4). A TCID50 assay was performed to quantify viral infections based on the cytopathic effect of the virus on VeroE6 cells in the presence and absence of increasing concentrations of PDC-109. This assay is rather crude and should be considered as a more qualitative rather than a quantitative measure of virus infection. Nonetheless, we found again that PDC-109 inhibits SARS-CoV-2 levels, albeit only to roughly 50% at the maximal concentration, but without any significant viability effects (Appendix A). As outlined in the results section, our assay likely underestimates the protective effects of PDC-109 since only the first two hours of infection are blocked in the treated cells. All secondary infections that occur in the subsequent 70 h of infection are completely PDC-109-independent, which may partly overwrite the initial effects of the treatment. In order to circumvent this shortcoming, we performed a complementary experiment in which the cells were infected for 24 h only, thereby effectively abolishing secondary infections. Expectedly, SARS-CoV-2 infections were almost completely suppressed at high PDC-109 concentrations, demonstrating its strong antiviral effects again.

Finally, we have assessed the antiviral activity of PDC-109 against VSV* reporter viruses pseudotyped with VSVg (Figure 5). VSV utilizes the ubiquitously expressed LDL-receptor (LDL-R) [47] for viral entry, thus enabling VSVg-mediated entry into a broad spectrum of mammalian cell lines and primary cell lineages. In this experiment, we included dSP derived from ejaculates of domestic cattle (*Bos taurus*), containing PDC-109, but also a multitude of other proteins and components, which might either facilitate or suppress antiviral properties of PDC-109 or even have independent inhibitory effects. Again, PDC-109 significantly and dose-dependently inhibited VSV*VSVg infections of VeroE6 cells, and dSP showed a comparable effect, suggesting that PDC-109 is the dominant antiviral factor in SP in our experimental setup. Noteworthily, a recent study has shown strong antiviral effects of human SP against Mumps virus infections, and a non-protein component was identified as the bioactive agent [3]. However, to the best of our knowledge, bull SP has hitherto not been thoroughly assessed for antiviral properties, nor have individual factors been identified with protective functions against invading pathogens.

Our results raise the question of how PDC-109 exerts its antiviral activity, but also the cytotoxicity we found, extended exposure. Previous studies have already shown that PDC-109 is able to lyse human red blood cells, and the extent of that depends on the cholesterol content in the erythrocyte membrane. We found a strong decrease in cell viability when incubating the cells with high concentrations of PDC-109 for long time intervals (above 22 h), whereas when treating the cells for 2 h with PDC-109, the cell viability was not significantly influenced. The later result argues against a sole impact of PDC-109 on membrane lipid composition and/or physical–chemical membrane properties as a cause of cell lysis (see below) since these modifications should act on a shorter time scale. Therefore, other explanations, possibly also considering intracellular effects of PDC-109, have to be taken into account (see also below). However, at the current state, the reasons for the disruption of the cells in the presence of PDC-109 are unknown and will be investigated in future studies.

With regard to the specific molecular mechanisms responsible for the observed antiviral effects, several possibilities seem conceivable. It could be hypothesized that PDC-109 binds to the viral and/or to the plasma membrane (of the target cells), thereby suppressing the attachment of viruses to the cell membrane. Recently, we proposed that such a lipid-binding is mainly responsible for the inhibition of influenza-mediated fusion in the presence of PDC-109 [14]. This mechanism would require that PDC-109 is present during the initial virus-plasma membrane interaction. Indeed, we found that PDC-109 was inhibitory when it was added to cells simultaneously with the virus, but not upon a pre-treatment. An interaction of PDC-109 with viral and/or cell membranes can be explained by its affinity to phosphorylcholine-containing lipids present in either membrane [22]. Alternatively, PDC-109 may also specifically interact with viruses via its Fn II domain. Several studies have shown that infection by many pathogenic bacteria and viruses involves the interaction of their surface protein with the Fn II domain of fibronectin present on the host [48,49,50,51,52,53].

It is also plausible that PDC-109 influences virus infectivity by more complex mechanisms that would not require PDC-109 to be physically present at the cell–virus interface. Indeed, our experiments on Calu6 ACE2 cells, in which PDC-109 almost completely abolished infection upon administration after completion of virus entry (Figure 2), indicate effects on virus replication and other intracellular processes. This could suggest that PDC-109 triggers cellular signaling cascades upon binding of cell surface components, which in turn impact viral post-entry steps. Otherwise, PDC-109 may also enter treated cells by endocytosis and then directly control virus infections intracellularly. Our finding that cytotoxicity of PDC-109 can be prevented by the removal of cellular supernatants argues against such large-scale, active internalization of PDC-109.

Nonetheless, based on a large body of literature, we surmise that membrane interactions are critically involved in the antiviral activity of SP and PDC-109. In that context, several known effects of PDC-109 have to be considered: (i) incorporation into the membrane bilayer (e.g., influencing membrane curvature and mobility of membrane lipids) [23,24,54,55], (ii) specific interaction with phosphorylcholine-containing lipids and cholesterol (which may modify lateral membrane organization) [22,27,56], and (iii) extraction of phospholipids, and preferentially, cholesterol from the plasma membrane (e.g., modifying lipid composition) [57,58]. These parameters and properties are also known to be important for the extent of virus infectivity. For instance, numerous studies have shown the role of the lipid composition of the target and of the virus membrane, especially the presence and the concentration of cholesterol and sphingolipids [59,60,61]. For cholesterol, an influence can be mediated by a direct interaction via sterol-binding proteins or by its property to form lateral membrane domains [62,63,64]. Those membrane changes may again indirectly influence membrane proteins that are involved during virus infection [64]. Moreover, the fusion of viruses with target membranes depends on the surface curvature of the participating membranes, which is influenced again by cholesterol or by the insertion of peptides/proteins [65,66,67,68,69]. Additionally to these lipid-specific effects of PDC-109, also a direct influence on proteins has to be considered based on the described chaperone-like activity [25], i.e., it may interact with proteins that are important for virus replication, e.g., transcription factors, by that exerting an antiviral activity.

## 5. Conclusions

Our study has shown, for the first time, significant pan-viral inhibitory effects of the bull SP protein PDC-109 in different cell culture systems and against different viral pseudotypes. PDC-109 potently blocks virus entry and unexpectedly also VSV* virus replication. However, it does not provide lasting protection from viral infections. Upon extended exposure, PDC-109 has significant cytotoxic effects, which can be completely mitigated by early removal and limited application during receptor binding and viral entry only. Our results may represent a novel, interesting direction for the design and development of new antiviral and antibacterial drugs since SP components from industrial farm animals and livestock are available in large quantities or, in the case of proteins, can be recombinantly produced [26] with the aim of a medical application if beneficial effects are identified.

## Figures and Tables

**Figure 1 viruses-14-02031-f001:**
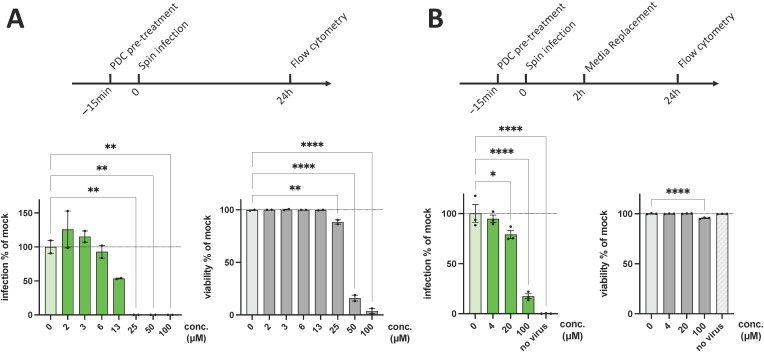
Anti-SARS-CoV-2 activity of PDC-109. (**A**) Calu6 ACE2 cells were pre-treated with different concentrations PDC-109 for 15 min and spin-infected with CoV-2 spike pseudoviruses (VSV*SARS CoV-2). Then, cells were incubated with virus and PDC-109 for 24 h and subjected to flow cytometry. (**B**) Calu6 ACE2 cells were pre-treated with different concentrations of PDC for 15 min and spin-infected with CoV-2 spike pseudoviruses in presence of PDC. Subsequently, excess virus and PDC were removed after 2 h of incubation, and cells were cultured for another 24 h. In both experiments, reporter virus signal and cell viability were assessed by flow cytometry. Frequencies of GFP+ cells are shown as a measure of VSV*SARS CoV-2 infection and were normalized to mock-treated samples (0 μM). Significance was assessed by parametric one-way analysis of variance (ANOVA) tests, comparing all samples with the mock control and displayed as follows: **** *p* < 0.0001; *** *p* < 0.001; ** *p* = 0.001–0.01; * *p* = 0.01–0.05. IC50 and CC50 values, as well as the selective index, can be found in Appendix A.

**Figure 2 viruses-14-02031-f002:**
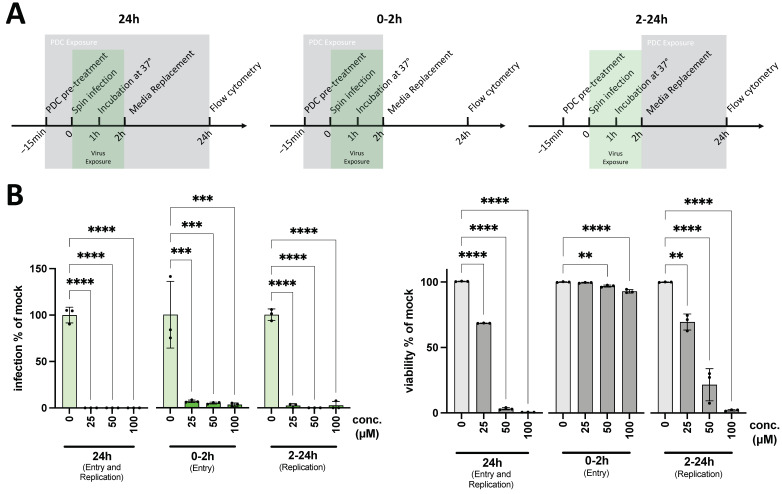
PDC-109 effects on virus entry and replication. (**A**) Experimental workflow for time-of-addition assay. Calu6 ACE2 cells were exposed to various concentrations of PDC at different phases of infection. The 24 h samples were incubated with PDC (grey box) for the entire infection period (green box), thus blocking virus entry and reporter virus infection. The 0–2 h samples were incubated with PDC-109 for the initial 2 h of infection only, limiting direct antiviral effects predominantly to the entry stage of infection. 2–24 h samples were treated with PDC at 2 h post-infection and, therefore, after successful viral entry. (**B**) Infection and viability were assessed by flow cytometry at 24 h post-infection. Frequencies of GFP+ cells are shown as a measure of VSV*SARS CoV-2 infection and were normalized to mock-treated samples (0 μM). Labels indicate PDC-109 concentration in μM. Significance was assessed by parametric one-way analysis of variance (ANOVA) tests, comparing all samples with the mock control (0 μM) and displayed as follows: **** *p* < 0.0001; *** *p* < 0.001; ** *p* = 0.001–0.01; * *p* = 0.01–0.05. IC50 and CC50 values as well as the selective index, can be found in Appendix A.

**Figure 3 viruses-14-02031-f003:**
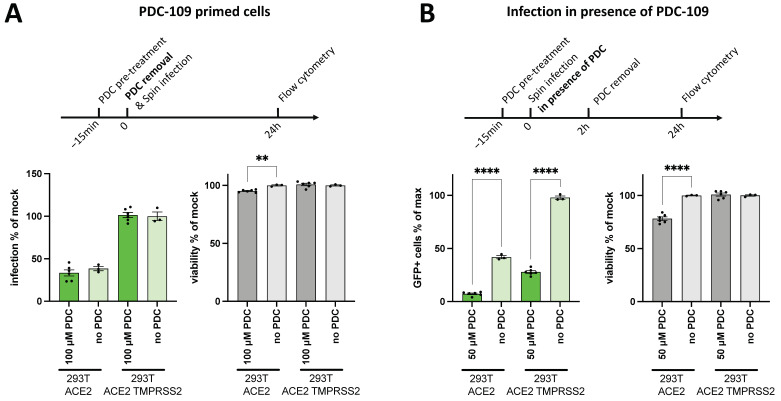
Permissivity of PDC-109 pre-treated cells. (**A**) 293T ACE2 and 293T ACE2 TMPRSS2 cells were pre-treated with 100 μM PDC-109 for 15 min and spin-infected with CoV-2 spike pseudoviruses in absence of PDC. (**B**) 293T ACE2 and 293T ACE2 TMPRSS2 cells were pre-treated with 50 μM PDC-109 for 15 min and spin-infected with CoV-2 spike pseudoviruses in presence of PDC for 30 min, followed by 90 min incubation at 37 °C. Subsequently, cells were washed and cultured further. Reporter virus signal and cell viability were assessed by flow cytometry at 24 h p.i. Frequencies of GFP+ cells are shown as a measure of VSV*SARS CoV-2 infection and were normalized to mock-treated samples (0 μM). Significance was assessed by parametric one-way analysis of variance (ANOVA) tests, comparing all samples with their respective mock control (no PDC) and displayed as follows: **** *p* < 0.0001; *** *p* < 0.001; ** *p* = 0.001–0.01; * *p* = 0.01–0.05.

**Figure 4 viruses-14-02031-f004:**
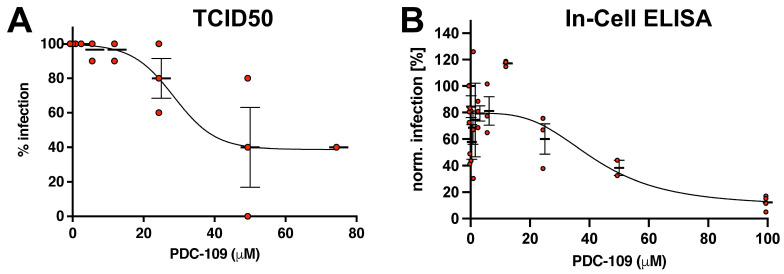
PDC-109 effects on live CoV-2 infection in cell culture. The antiviral activity of PDC was assessed by TCID50 assay or In-Cell ELISA. (**A**) VeroE6 cells were pre-treated with different concentrations of PDC in 10 repeats for 15 min and incubated with infectious CoV-2 (Essen isolate) for 2 h at 37 °C. Subsequently, PDC-containing media were replaced with fresh media, and cells were incubated for another 3 to 5 days to allow for spreading infections in the cell culture. Trypan blue staining was conducted to determine viral cytopathic effects. The frequency of wells with intact cell monolayers per PDC concentration was utilized as a measure of viral infectivity. Each point represents the data of one independent experiment. Solid lines show inhibitor-response curves generated using four-parameter fits. (**B**) VeroE6 cells were pre-treated with different concentrations of PDC and incubated with infectious CoV-2 (Essen isolate) for 2 h at 37 °C, followed by media exchange and incubation at 37 °C. In-Cell ELISA was performed 24 h post-infection. Dots show the mean of three technical repeats per condition from 2–3 independent experiments. Thick lines show arithmetic mean with SEM. B shows data normalized to mock-treated infection controls. IC50 values can be found in Appendix A.

**Figure 5 viruses-14-02031-f005:**
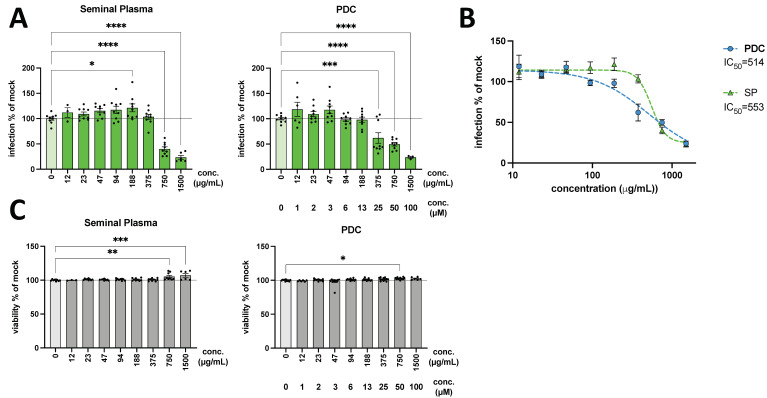
Activity of PDC-109 and SP against VSVg pseudotype infections. VeroE6 cells were pre-treated for 15 min with SP or PDC-109, respectively. Then, cells were spin-infected in presence of both reagents for 30 min, followed by 90 min incubation at 37 °C. Subsequently, cells were washed and cultured for another 24 h. (**A**) The frequency of GFP+ VeroE6 cells, reflecting the extent of infectivity, was assessed for SP and PDC-109 (PDC) based on flow cytometry measurements. Data were normalized to mock-treated control infections. Bars show mean with SEM. Significance was assessed by parametric one-way analysis of variance (ANOVA) tests, comparing all samples with the mock control (0 μM) and displayed as follows: **** *p* < 0.0001; *** *p* < 0.001; ** *p* = 0.001–0.01; * *p* = 0.01–0.05. Normality of the data was assessed with a Shapiro–Wilk test. (**B**) Inhibitor-response curves were generated using four-parameter fits. IC_50_ values in μg/mL are shown in the legend. (**C**) Viability was assessed by flow cytometry and normalized to mock-treated controls. IC50 values can be found in Appendix A.

## Data Availability

Supplementary data are provided. Raw data are available upon request.

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
