# Peer review of "The Bovine Seminal Plasma Protein PDC-109 Possesses Pan-Antiviral Activity"

_viruses, 2022, doi:10.3390/v14092031_

Round 1

Reviewer 1 Report (Previous Reviewer 1)

The authors reasonably addressed my specific comments and resolved my main doubts. I would like to thank for their clarifications, modifications made to the manuscript, as well as their point-by-point answers to each of the questions raised. I do recommend the manuscript for publication.

Author Response

We thank the reviewer for his service and positive response.

Reviewer 2 Report (Previous Reviewer 2)

The manuscript has been revised as to cope with most of the initial comments. In my opinion, there is one point that still needs improvement. Table S1 shows little information, due to the impossibility of calculating a precise value for some parameters. We suggest that when CC50 cannot be calculated because it is too high, data could be expressed, referred to the highest concentration tested: for example, IC50= 10 mM, CC50>100 mM and SI >10. By expressing the data in this way, table S1 will become much more informative.

Author Response

We thank the reviewer for this excellent suggestion. The SI have been revised accordingly (see Table S1).

This manuscript is a resubmission of an earlier submission. The following is a list of the peer review reports and author responses from that submission.

Round 1

Reviewer 1 Report

The antiviral activity of PDC-109 protein (one of the major soluble component of bovine seminal plasma) against Severe Acute Respiratory Syndrome Coronavirus-2 (SARS-CoV-2) has been analyzed in this work by Schwarzer-Sperber et al. The function of seminal plasma as antimicrobial and antiviral activity against some viral pathogens has been previously described (as mentioned in the manuscript), as well as the inhibitory effect of PDC-109 protein against Influenza virus (IAV). In this work, the authors describe a significant antiviral effect of PDC-109 protein on reporter VSV expressing spike protein of SARS-CoV-2 or glycoprotein of VSV when cells are pre-treated with a dose less than the cytotoxic dose administered in transient. This observation allows them to conclude that the PDC-109 protein has broad activity against viral pathogens due to its antiviral effect exerted on both non-related viruses by an unknown mechanism.

Although the results are interesting, some of them have clear limitations that I think should be addressed by the authors before publication. Below, I list some suggestions that I hope will help strengthen the manuscript.

General comments.

- The information  provided in the M&M section is poorly detailed for easy reading. This section needs to be revised and completed with details on experiments that are commented throughout manuscript, but how they were done is not detailed. The footnotes of figures is sparse and difficult to read and relate to the results shown in the panels of each figure. 

- Figures are not referenced in the text or appear as an error. 

- Bibliographic citations must be correctly indicated in the "Results" section. Many references are missing in the text.

- Errors and reference sources not found in the manuscript require correction. The results are difficult to understand for this reason too. 

Major

- Based on the antiviral activity of PDC-109 detected against IAV (through a previously well-characterized mechanism), the authors decide to test the antiviral activity of PDC against two non-related virus species (SARS-CoV-2 and VSV) and to extend the antiviral effect of this protein, Precisely, this is one of the main criticisms of this work. The authors analyze the effect of PDC-109 protein against SARS-CoV-2 and VSV, when it really comes down to analyzing the effect only for VSV. Except one (see Fig. 4), all experiments are performed using VSV as a reporter virus harboring the SARS-CoV-2 spike protein and vesicular stomatitis virus glycoprotein. 

- The absence of physiological experiments in the work is very clear, which together with the scarce information provided by the authors hinder its evaluation as pan-antiviral.

- It ‘s necessary to describe what the VSV*SARS-CoV 2 pseudotype used in the different experiments to understand the conclusion and results. If this has previously described, the authors could give the reference to undestand what is base on. 

- In infection experiments. Why a pre-treatment with PDC-109 is necessary before infection in culture cells?. Is the level of viral entry and replication altered compared to non-pretreated cells? What happens if they are not pre-treated?.

- Anti-SARS-CoV-2 activity effect of PDC-109 is analyzed in Calu 6 cells in which ACE2 and/or TMPRSS2 are stably overexpressed by transduction using lentiviral particles containing each of the aforementioned receptors. Have the authors considered carrying out these same experiments with more physiological cells? eg. Calu3-2B4 cells, a biological clone derived from Calu3 cells, overexpress ACE2 and TMPRSS2 receptors and making them much more susceptible to SARS-CoV-2 infection than the parental Calu3 cells. The differences between Calu3 and Calu3-2B4 could give interesting results to allow a better evaluation of the antiviral activity potential of PDC-109 against live SARS-CoV-2 virus.

- Fig. 2 B. Given that cell viability appears null when using PDC-109 concentrations greater than 25-50 nM, could a lower range not have been studied? This could have been more informative. 

- How do the authors explain the differences in cell viability found in different experiments when cells are treated with the same concentration of PDC-109 protein (50µM)? Compare results of Fig. 3, B and Fig. 2, B (24h).

- The antiviral activity effect of PDC-109 has been so far described  against enveloped viruses. Have the authors attempted to analyze the inhibition of PDC-109 or bovine seminal plasma against non-enveloped viruses? Based on the action mechanism of PDC-109, in which this protein binds to the cells membranes preventing attachment of the virus to the cell and reducing the efficiency of membrane fusión, is it possible that a non-enveloped virus is not affected? 

- With the aim to determine the inhibitory potential of PDC-109 in cell cultures of the same species as bovine protein, Have the authors tried to perform a classical antiviral activity assay with PDC-109 in bovine cultured cells? (e.g. MDBK cells). The bovine cells could be treated with PDC-109 protein or seminal plasma and then infected with VSV. The supernatant of infection collected could be then titered in susceptible culture cells. 

- Some parameter indicative of the potency of PDC-109 on the viral inhibitions shown could be provided in M&M  and commented in the manuscript (eg, half maximal inhibitory concentration, IC50), half citotoxicity index (CC50) or selectivity index (SI) as a ratio to relate cytotoxicity and antiviral activity as a parameter to express the efficacy of PDC-109 protein in vitro on inhibiting virus replication.

- The resolution of the images shown in Fig. S2, panel B does not appear to show cell viability, however, the cell viability graph shown in panel C (related to the effect of PDC-109 under the experimental conditions indicated in B) shows complete or close to 100% viability. Is this correct?

Minor

- the sentence in the introduction section (pag. 2, lin-72-73: "...with only minor eukaryotic cytotoxicity after transient protein administration") could be better explained for better understanding. 

- The level of statistical significance of the analyzed data is missing. In case of being statistically significant, it should be indicated with different asterisks in the figures and graphs, as mentioned in M&M.

- The multiplicity of infection (moi), regardless of the virus used in the experiments could be indicated in the corresponding section.

- What does spin infection mean?.  (Fig. 1, 2 and 3) Could the authors clarify their meaning in the M&M? 

- Pag 5, lin. 221. “… at double digit micromolar concentrations.” For a better understanding, this phrase could be replaced by the concentration range in which the authors claim to inhibit infection. 

- The word "protein" appears many times throughout the manuscript without specifying who it is, it is probably associated with PDC-109, but it is not indicated. (p.e.pag.5, lin.222, lin.238, lin.240;  pag10, lin.406, lin.407, lin.419; pag.11, lin.429, lin.440). Could the authors specify which protein they are referring to? 

- Pag.9, lin.356. “… faithfully replicate the entry-requirements and other pathogen-specific cellular-processes of the wildtype viruses the pseudotyping protein was derived of [35–40]”. Is it correct how the sentence ends?

- Pag 7, lin. 284. What does “in cellulo” mean in a viral infection context?.  

- Panel B in Figure 5 is not shown, however information about it is indicated in the footnote.

- The title of the reference 31 (Hoffmann et al (2020) Cell 181, 1–10) is duplicated. This could be modified.

Reviewer 2 Report

The authors have observed an antiviral activity of the seminal plasma protein PDC-109 against reporter viruses, pseudotyped with the spike protein of SARS-CoV-2 or the glycoprotein of VSV, upon transient administration. Limited results are also shown on the ability of PDC-109 to directly inhibit SARS-CoV-2 virus infection.  

Overall, the topic is interesting and the results promising but there is a lack of relevant information, and some conflicting results. Therefore, I recommend resubmission of  a new version of this manuscript addressing the following issues:

 - Direct virucidal effect of the PDC-109 on virus particles has not been shown. These experiments are crucial before testing drugs for their possible antiviral effect, and relevant to understand the antiviral activity observed.

- There are no data of the M.O.I in any of the infection assays presented.

- Results are interpreted as “strong inhibition” or “significant cytotoxicity” but the data on both inhibition (IC50) and cytotoxicity (CC50) are missing in almost all trials (Figs 1, 2, 3 and 4). These parameters are essential for antiviral compound evaluation, including the estimation of the selectivity index (SI).

- PDC-109 pretreatment enables VSV*SARS-CoV-2 inhibition without affecting cell viability in Calu6 ACE2 cells, but not in 293T ACE2 nor in 293T ACE2 TMPRSS2 cells. Could you discuss these conflicting result? When studying cell-targeted antivirals it is important to establish a standard concentration and administration protocol that ensures the antiviral effect without affecting cell viability (SI).

 - Figure 4A. SEM between experiments are too high to conclude that inhibition levels reach 50%, on the other hand a low value considering potential antiviral applications.

-        - Figure 5. Testing of higher concentrations of SP and PDC can result in greater inhibitions.